# Risk Factors and Clinical Characteristics of Acute Kidney Injury in Patients with COVID-19: A Systematic Review and Meta-Analysis

**Amal Arifi Hidayat** [1], **Vania Azalia Gunawan** [1], **Firda Rachmawati Iragama** [1], **Rizky Alfiansyah** [1], **Decsa Medika Hertanto** [2], **Artaria Tjempakasari** [2,*] and **Mochammad Thaha** [2]

[1] Internal Medicine Resident, Department of Internal Medicine, Dr. Soetomo Hospital, Faculty of Medicine, Universitas Airlangga, Surabaya 60132, Indonesia; arifiamal@gmail.com (A.A.H.); vaniazalia9@gmail.com (V.A.G.); firwessels@gmail.com (F.R.I.)

[2] Division of Nephrology–Hypertension, Department of Internal Medicine, Dr. Soetomo Hospital, Faculty of Medicine, Universitas Airlangga, Surabaya 60132, Indonesia; decsa.medika.hertanto@gmail.com (D.M.H.); thaha.505.mt@gmail.com (M.T.)

[*] Correspondence: artnef@yahoo.com

**Abstract:** Acute kidney injury (AKI) is associated with a worse prognosis in coronavirus disease 2019 (COVID-19) patients. Identification of AKI, particularly in COVID-19 patients, is important for improving patients' management. The study aims to assess risk factors and comorbidities of AKI in COVID-19 patients. We systematically searched PubMed and DOAJ databases for relevant studies involving confirmed COVID-19 patients with data on risk factors and comorbidities of AKI. The risk factors and comorbidities were compared between AKI and non-AKI patients. A total of 30 studies involving 22385 confirmed COVID-19 patients were included. Male (OR: 1.74 (1.47, 2.05)), diabetes (OR: 1.65 (1.54, 1.76)), hypertension (OR: 1.82 (1.12, 2.95)), ischemic cardiac disease (OR: 1.70 (1.48, 1.95)), heart failure (OR: 2.29 (2.01, 2.59)), chronic kidney disease (CKD) (OR: 3.24 (2.20, 4.79)), chronic obstructive pulmonary disease (COPD) (OR: 1.86 (1.35, 2.57)), peripheral vascular disease (OR: 2.34 (1.20, 4.56)), and history of nonsteroidal anti-inflammatory drugs (NSAID) (OR: 1.59 (1.29, 1.98)) were independent risk factors associated with COVID-19 patients with AKI. Patients with AKI presented with proteinuria (OR: 3.31 (2.59, 4.23)), hematuria (OR: 3.25 (2.59, 4.08)), and invasive mechanical ventilation (OR: 13.88 (8.23, 23.40)). For COVID-19 patients, male gender, diabetes, hypertension, ischemic cardiac disease, heart failure, CKD, COPD, peripheral vascular disease, and history of use of NSAIDs are associated with a higher risk of AKI.

**Keywords:** 2019-nCoV disease; acute kidney injury; COVID-19; SARS-CoV-2 infection

## 1. Introduction

The emergence of coronavirus disease 2019 (COVID-19), caused by severe acute respiratory syndrome coronavirus 2 (SARS-CoV-2), infected over 40 million people and was declared a pandemic on 11 March 2020, by the World Health Organization (WHO) [1]. The majority of COVID-19 patients were asymptomatic or presented with mild upper respiratory illness symptoms. However, COVID-19 can lead to the development of a more severe condition, leading to death [2]. As of 29 October 2020, there have been 44,774,763 confirmed cases of COVID-19, causing 1,179,225 deaths worldwide [3].

The previous meta-analysis revealed that acute kidney injury (AKI) was associated with a worse prognosis in COVID-19 patients [4]. AKI developed in 5–15% of COVID-19 patients and carried a high mortality rate [2–5]. Patients who recovered from AKI were still independently associated with long-term mortality, cardiovascular events, and the development of chronic kidney disease (CKD) [6]. Several risk factors, including advanced age, male gender, cardiovascular disease, hypertension, diabetes, chronic kidney disease

(CKD), and chronic liver disease, were reported to be at higher risk for SARS-CoV-2 infection with a poor prognosis [7,8]. Even though the relationship between AKI and severity or mortality has been established, the risk factors and comorbidities of AKI in COVID-19 patients remain unclear. Hence, we performed a systematic review with meta-analysis aiming to provide information about risk factors and comorbidities of AKI, particularly in COVID-19 patients.

## 2. Materials and Methods

### 2.1. Search Strategy and Selection Criteria

We conducted a systematic search on PubMed and DOAJ databases from inception to 10 October 2020, for articles using the keywords "COVID-19", "2019-nCoV disease", "SARS-CoV-2 infection" in combination with "acute kidney injury", "acute kidney failures", "acute renal failures", "acute renal injury". Additional studies were retrieved by manual screening of the reference lists of other meta-analyses and systematic reviews identified. The search was limited to original research with full text available in English. AKI was defined according to the 2012 KDIGO definition, which is an increase in serum creatinine $\geq 0.3$ within 48 h, or an increase in serum creatinine $\geq 1.5$ times baselines within 7 days, or a decrease in urine volume $< 0.5$ mL/kg/h for 6 h [9]. Prospective cohort, retrospective cohort, case–control, and cross-sectional studies involving adult patients with COVID-19, containing data on risk factors and comorbidities of AKI, met the inclusion criteria.

The following studies were excluded: duplicate publications, reviews, editorials, case series, case reports, guidelines, basic research, preprint papers, studies that do not provide risk factors or comorbidities data, and studies that include children as subjects. Four reviewers (A.A.H., V.A.G., F.R.I., and R.A.) independently screened the titles and abstracts for articles and read the full text for the potentially eligible studies. Conflicts were discussed and resolved by the third, fourth, and fifth senior reviewers (D.M.H., A.T., and M.T.). The selection process is shown with a flow chart (Figure 1).

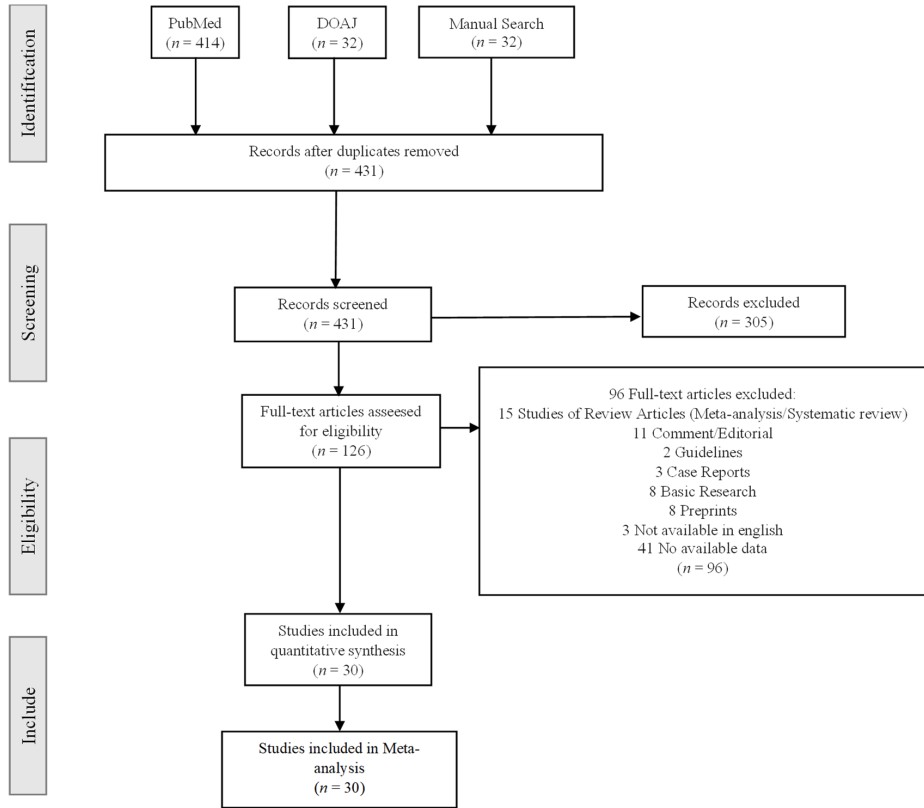

**Figure 1.** Selection process on studies.

## 2.2. Data Extraction and Quality Assessment

Relevant information including the first author, year of publication, study design, location, age, gender, number of populations, AKI prevalence, mortality rate, and proportion of patients requiring renal replacement therapy (RRT) was identified and extracted. Four reviewers (A.A.H., V.A.G., F.R.I., and R.A.) independently extracted data on risk factors and comorbidities, such as the prevalence of males, advanced age, diabetes mellitus, hypertension, obesity, smokers, ischemic cardiac disease, heart failure, cerebrovascular disease, chronic kidney disease (CKD), chronic obstructive pulmonary disease (COPD), peripheral vascular disease, chronic liver disease, cancer, and HIV (human immunodeficiency virus) status. A history of chronic use of drugs was also retrieved. The available characteristics were extracted, including laboratory results and concomitant treatments.

The risk of bias was assessed by NIH quality assessment tools. Disagreement was resolved by discussion with senior reviewers (D.M.H., A.T., and M.T.) until a consensus was reached. The third and fourth authors discussed and resolved differences of opinion.

## 2.3. Data Synthesis and Statistical Analysis

The odds ratio (OR) was used to describe the ratio of the probability of events occurring in acute kidney injury patients (AKI) versus non-acute kidney injury patients (NAKI). Forest plots were created to present the prevalence and the corresponding 95% CI of risk factors and comorbidities. We used the $I^2$ statistic to assess heterogeneity among the studies. $I^2$ values from 0% to 50% indicate low heterogeneity, $I^2$ values between 50% and 75% indicate moderate heterogeneity, and $I^2$ values greater than 75% indicate high heterogeneity. If $I^2 < 50\%$, we used the fixed benefit model to pool the data. Contrarily, when $I^2 > 50\%$, we used the random-effect model. The threshold of statistical significance was set to 0.05. We used a funnel plot to test publication bias. All analyses and plots were performed and created with Review Manager (version 5.3).

## 3. Results

### 3.1. Search Results, Characteristics of the Included Studies, and Methodological Quality

Figure 1 describes the step-by-step method we followed to include studies in our meta-analysis. We initially identified 478 articles in our research. After removing the duplicates, 431 articles remained. From titles and abstract screening, 305 were excluded. A total of 126 potentially eligible articles were assessed by full-text review. Of these 126 studies, 96 articles met the exclusion criteria. A total of 30 studies fulfilled our selection criteria and included the data we need to investigate. A total of 22,385 patients were included in these studies. The main characteristics of patients and these studies are described in Table 1.

**Table 1.** The main characteristics of included studies.

| First Author | Published Year | Study Design | Location | Number of Populations | Age | Gender (Male%) | AKI | N (%) Mortality | RRT |
|---|---|---|---|---|---|---|---|---|---|
| Alberici [10] | 2020 | Cohort | Italy | 20 | 57 (44–70) | 16 (80) | 5 (25) | 5 (25) | 9 (17.3) |
| Chan [11] | 2020 | Cohort | USA | 3993 | 64 (56–78) | 2289 (57.32) | 2158 (46) | / | 347 (18.9) |
| Chand [12] | 2020 | Cohort | USA | 300 | 58.2 (12.6) | 182 (60.7) | 230 (76.7) | 157 (52.3) | 117 (39) |
| Cui [13] | 2020 | Cohort | China | 116 | 61.05 (12.9) | 66 (56.89) | 21 (18.1) | 24 (20.6) | / |
| Duduignon [14] | 2020 | Cross Sectional | France | 51 | 63 (57–69) | 39 (76.5) | 26 (50.98) | 14 (27.4) | 10 (39) |
| Fisher [15] | 2020 | Cohort | USA | 3345 | 64.4 (16.4) | 1776 (53.1) | 1903 (56.9) | 775 (23.2) | 574 (17.1) |
| Forminsky [16] | 2020 | Cohort | Italy | 96 | 64 (58.5–70.0) | 80 (83.33) | 72 (75) | 32 (33.33) | 17 (17.70) |
| Grimaldi [17] | 2020 | Cohort | Europe | 414 | 63 (11) | 320 (77.2) | 231 (55.8) | 95 (22.9) | / |
| Hirsch [18] | 2020 | Cohort | USA | 5449 | 64 (52–75) | 3317 (60.9) | 1993 (36.6) | 888 (54.9) | 285 (5.2) |
| Husain [19] | 2020 | Cohort | Europe | 23 | 60 (37–88) | 19 (82.6) | 12 (52.2) | 15 (65.2) | 3 (13) |
| Joseph [20] | 2020 | Cohort | French | 100 | 59 (53–67) | 70 (70) | 81 (81) | 29 (29) | 13 (13) |
| Lee [21] | 2020 | Cohort | USA | 1002 | 66 (53–76) | 619 (62) | 294 (29.3) | 172 (17) | / |

**Table 1.** *Cont.*

| First Author | Published Year | Study Design | Location | Number of Populations | Age | Gender (Male%) | AKI | N (%) Mortality | RRT |
|---|---|---|---|---|---|---|---|---|---|
| Lei [22] | 2020 | Cohort | China | 34 | 55 (43–63) | 14 (41.2) | 2 (5.9) | 7 (20.6) | / |
| Lim [23] | 2020 | Cross Sectional | Asia | 160 | 61 (24–98) | 106 (66.25) | 30 (18.8) | 37 (23.12) | 5 (3.1) |
| Liu [24] | 2020 | Cross Sectional | China | 1190 | 57 (47–67) | 635 (53.36) | 51 (4.3) | / | / |
| Mohammed [25] | 2020 | Cohort | Australia | 575 | 65 (36–96) | 356 (62) | 161 (28.0) | 200 (34.7) | 89 (19.4) |
| Naar [26] | 2020 | Cohort | USA | 206 | 60 (47–71) | 134 (65.1) | 148 (71.8) | / | / |
| Naarayan [27] | 2020 | Cohort | USA | 370 | 71 (59–82) | 207 (55.9) | 182 (49.1) | 150 (40.5) | / |
| Nakeshbandi [28] | 2020 | Cohort | USA | 504 | 68 (15) | 263 (52) | 95 (19) | 219 (43) | / |
| Pei [29] | 2020 | Cohort | China | 333 | 56.3 (13.4) | 182 (54.7) | 35 (10.5) | 29 (8.7) | 6 (1.8) |
| Pelayo [30] | 2020 | Cross Sectional | USA | 223 | 70.30 (12.80) | 115 (51.56) | 110 (49.3) | 44 (19.73) | 3 (1.3) |
| Russo [31] | 2020 | Cohort | Italy | 777 | 70 (16) | 458 (59) | 176 (22.65) | 273 (35.13) | 21 (2.7) |
| Soleimani [32] | 2020 | Cohort | Iran | 254 | 66.4 (12.9) | 149 (58.7) | 49 (19.3) | 68 (26.8) | / |
| Taher [33] | 2020 | Cohort | Asia | 73 | 54.3 (13.5) | 44 (60.3) | 29 (39.7) | 13 (17.8) | 7 (9.6) |
| Vee [34] | 2020 | Cross Sectional | Malaysia | 247 | 28 (18–35) | 172 (69.6) | 16 (6.5) | / | 7 (2.83) |
| Wang [35] | 2020 | Cohort | China | 116 | 62 (55–69) | 62 (53.4) | 12 (10.34) | / | / |
| Wu [36] | 2020 | Cohort | China | 1048 | 62.5 (48–77) | 591 (56.34) | 19 (1.8) | 52 (4.96) | 21 (2.00) |
| Xia [37] | 2020 | Cohort | China | 81 | 66.6 (11.4) | 54 (66.7) | 41 (50.6) | 60 (70.4) | 8 (9.9) |
| Yan [38] | 2020 | Cohort | China | 882 | 71 (68–77) | 440 (49.9) | 115 (13.0) | 128 (14.5) | 7 (1.9) |
| Zahid [39] | 2020 | Cohort | USA | 469 | 66 (55–75) | 268 (57.14) | 128 (27.29) | 188 (40.09) | / |

### 3.2. Data Synthesis

#### 3.2.1. Patient's Sex

Twenty-six studies of 20,363 patients reported the relationship between male gender and AKI in patients with COVID-19. The results obtained are shown in Figure 2. Due to the high level of heterogeneity, the random-effects model was selected. The meta-analysis demonstrated that male gender was associated with a higher risk of AKI (OR: 1.74 (1.47, 2.05), *p* < 0.05).

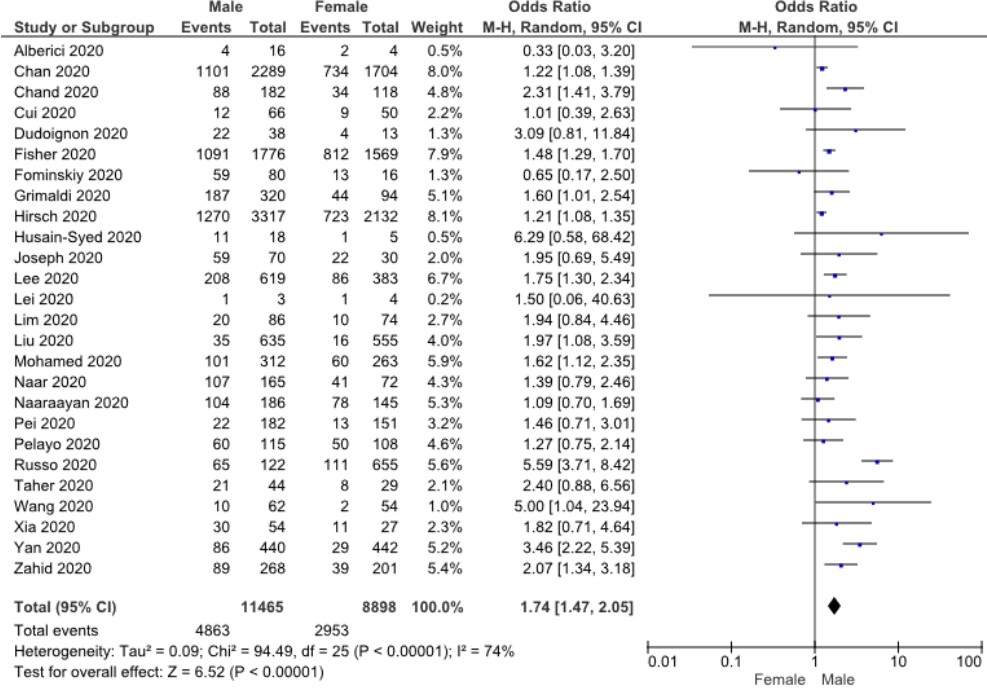

**Figure 2.** Forrest plot showing risk of AKI in COVID-19 male patients [10–27,29–31,33,35,37–39].

### 3.2.2. Patient's Comorbidity

A meta-analysis was performed to investigate the presence of comorbidities potentially associated with the risk of AKI. A total of 24 studies evaluated the role of diabetes, and 22 studies evaluated the role of hypertension in patients with COVID-19 and AKI. The heterogeneity was high in hypertension data; hence, the random-effects model was applied for these studies. Conversely, the fixed-benefit model was used in terms of diabetes due to its low heterogeneity. The meta-analysis showed that COVID-19 patients with diabetes (OR: 1.65 (1.54, 1.76), $p < 0.05$) (Figure 3a) and hypertension (OR: 1.82 (1.12, 2.95), $p < 0.05$) (Figure 3b) had a higher risk of AKI.

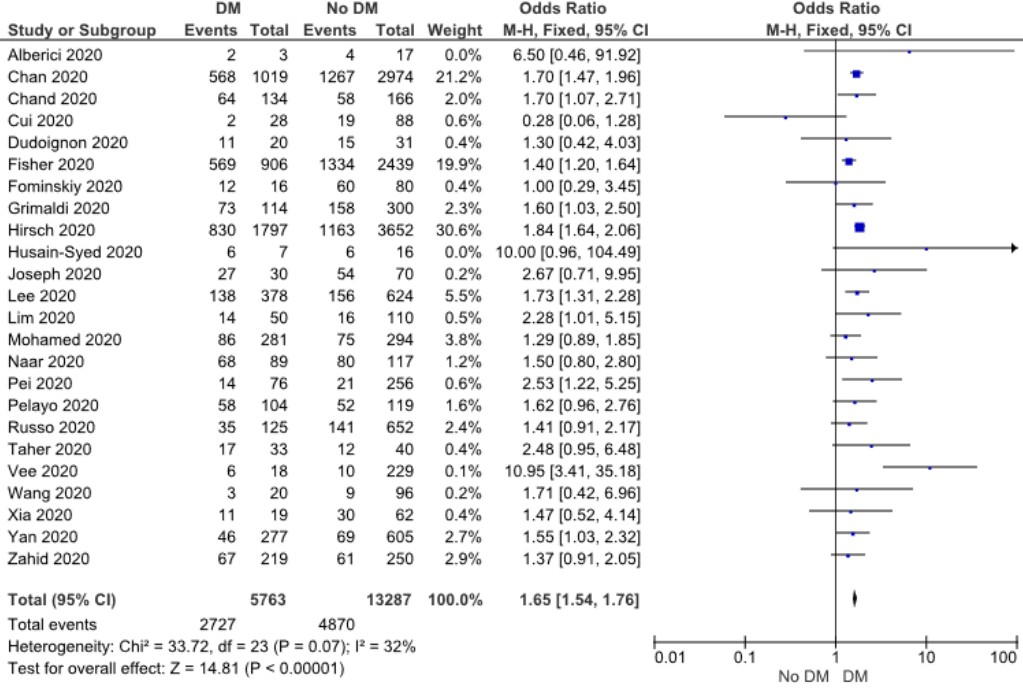

(**a**)

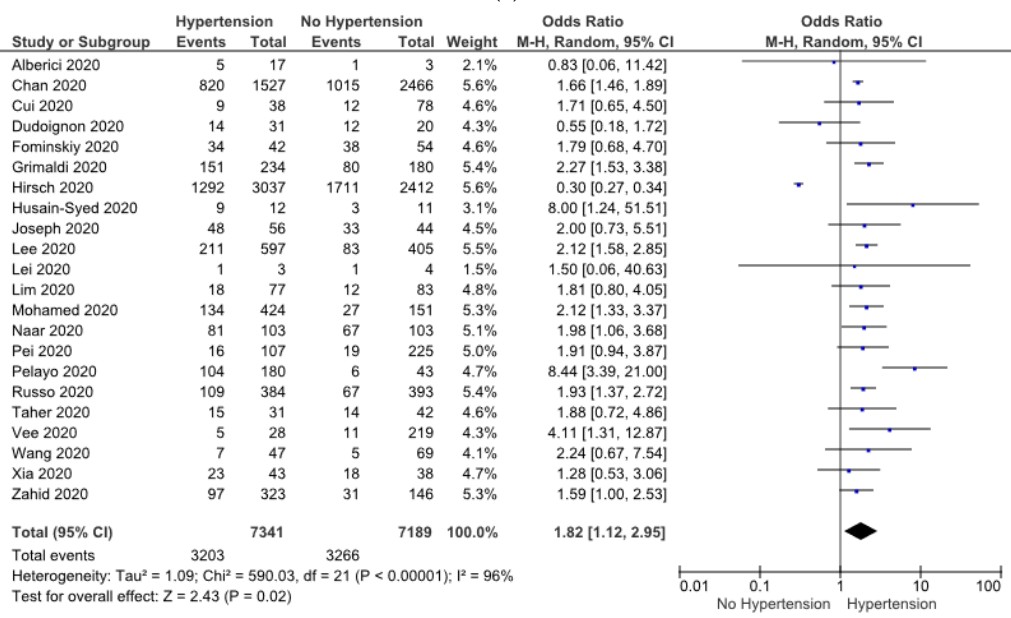

(**b**)

**Figure 3.** *Cont*.

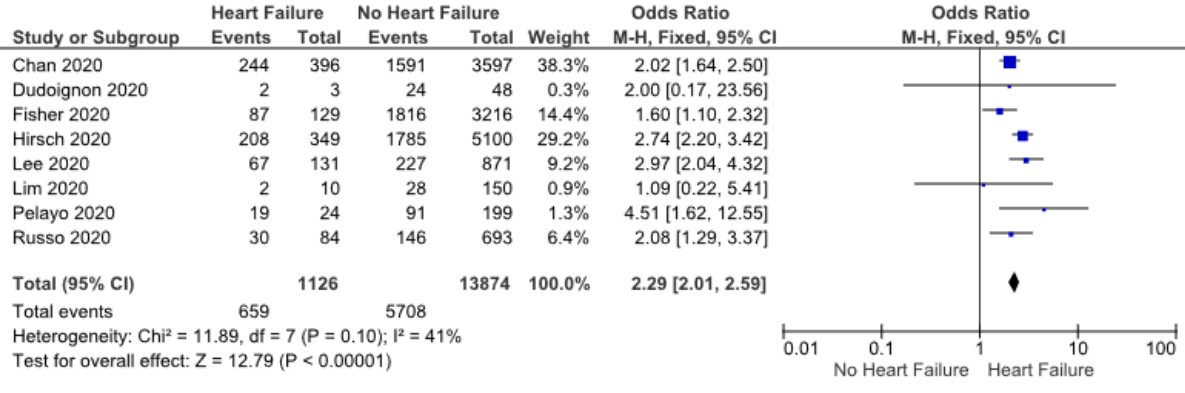

(**c**)

| Study or Subgroup | Ischemic cardiac disease Events | Total | No Ischemic cardiac disease Events | Total | Weight | Odds Ratio M-H, Fixed, 95% CI |
|---|---|---|---|---|---|---|
| Alberici 2020 | 1 | 3 | 5 | 17 | 0.3% | 1.20 [0.09, 16.44] |
| Cui 2020 | 10 | 48 | 11 | 68 | 2.5% | 1.36 [0.53, 3.52] |
| Fominskiy 2020 | 5 | 6 | 67 | 90 | 0.5% | 1.72 [0.19, 15.47] |
| Grimaldi 2020 | 27 | 37 | 204 | 377 | 3.4% | 2.29 [1.08, 4.86] |
| Hirsch 2020 | 289 | 600 | 1704 | 4849 | 66.5% | 1.72 [1.45, 2.03] |
| Husain-Syed 2020 | 3 | 5 | 9 | 18 | 0.5% | 1.50 [0.20, 11.24] |
| Lei 2020 | 1 | 4 | 1 | 3 | 0.3% | 0.67 [0.02, 18.06] |
| Lim 2020 | 5 | 21 | 25 | 139 | 1.7% | 1.43 [0.48, 4.25] |
| Naar 2020 | 15 | 19 | 133 | 187 | 1.8% | 1.52 [0.48, 4.80] |
| Pelayo 2020 | 20 | 35 | 90 | 188 | 4.1% | 1.45 [0.70, 3.01] |
| Russo 2020 | 30 | 90 | 146 | 687 | 7.7% | 1.85 [1.15, 2.98] |
| Taher 2020 | 6 | 9 | 23 | 64 | 0.6% | 3.57 [0.81, 15.61] |
| Wang 2020 | 3 | 12 | 9 | 104 | 0.5% | 3.52 [0.81, 15.37] |
| Xia 2020 | 8 | 17 | 33 | 64 | 2.5% | 0.84 [0.29, 2.44] |
| Zahid 2020 | 26 | 74 | 102 | 395 | 7.1% | 1.56 [0.92, 2.64] |
| **Total (95% CI)** | | **980** | | **7250** | **100.0%** | **1.70 [1.48, 1.95]** |
| Total events | 449 | | 2562 | | | |

Heterogeneity: Chi² = 5.35, df = 14 (P = 0.98); I² = 0%
Test for overall effect: Z = 7.43 (P < 0.00001)

(**d**)

| Study or Subgroup | CKD Events | Total | No CKD Events | Total | Weight | Odds Ratio M-H, Random, 95% CI |
|---|---|---|---|---|---|---|
| Chan 2020 | 339 | 420 | 1496 | 3573 | 10.6% | 5.81 [4.52, 7.47] |
| Cui 2020 | 1 | 5 | 20 | 111 | 2.4% | 1.14 [0.12, 10.73] |
| Fisher 2020 | 287 | 409 | 1616 | 2936 | 10.7% | 1.92 [1.54, 2.40] |
| Grimaldi 2020 | 28 | 33 | 203 | 381 | 6.6% | 4.91 [1.86, 12.99] |
| Joseph 2020 | 27 | 29 | 54 | 71 | 4.1% | 4.25 [0.91, 19.75] |
| Lee 2020 | 66 | 138 | 228 | 864 | 10.1% | 2.56 [1.77, 3.69] |
| Mohamed 2020 | 56 | 172 | 105 | 403 | 10.0% | 1.37 [0.93, 2.02] |
| Naar 2020 | 26 | 27 | 122 | 179 | 2.8% | 12.15 [1.61, 91.75] |
| Pelayo 2020 | 29 | 39 | 81 | 184 | 7.7% | 3.69 [1.70, 8.01] |
| Russo 2020 | 79 | 222 | 97 | 555 | 10.2% | 2.61 [1.84, 3.71] |
| Vee 2020 | 5 | 9 | 11 | 238 | 4.4% | 25.80 [6.07, 109.68] |
| Xia 2020 | 1 | 3 | 40 | 78 | 2.1% | 0.47 [0.04, 5.46] |
| Yan 2020 | 37 | 83 | 78 | 799 | 9.5% | 7.44 [4.55, 12.16] |
| Zahid 2020 | 18 | 50 | 110 | 419 | 8.7% | 1.58 [0.85, 2.93] |
| **Total (95% CI)** | | **1639** | | **10791** | **100.0%** | **3.24 [2.20, 4.79]** |
| Total events | 999 | | 4261 | | | |

Heterogeneity: Tau² = 0.36; Chi² = 90.50, df = 13 (P < 0.00001); I² = 86%
Test for overall effect: Z = 5.91 (P < 0.00001)

(**e**)

**Figure 3.** *Cont.*

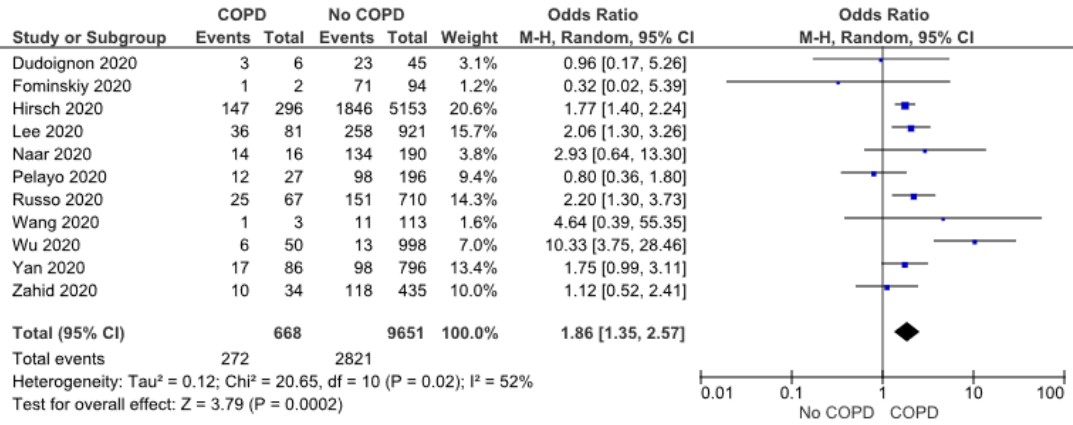

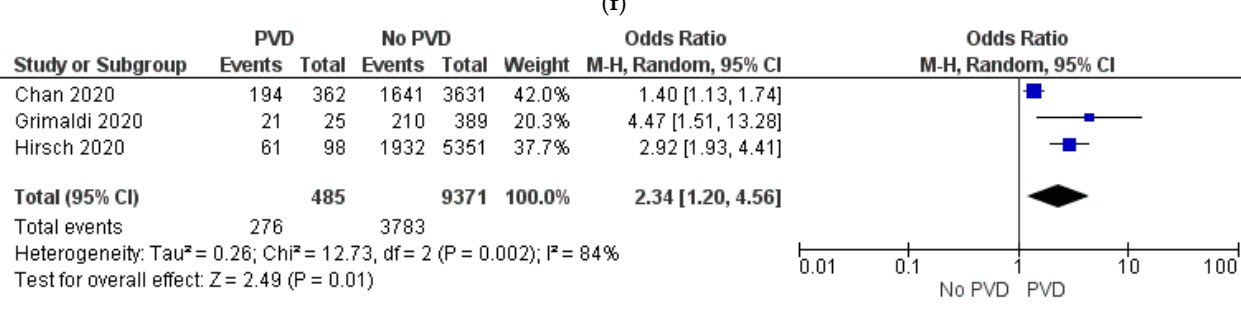

**Figure 3.** Forrest plot showing association between COVID-19 patients' comorbidities with risk of AKI. Diabetes (**a**), hypertension (**b**), heart failure (**c**), ischemic cardiac disease (**d**), CKD (**e**), COPD (**f**), and peripheral vascular disease (**g**) [10–23,25,26,29–31,33–35,37–39].

The meta-analysis revealed that heart failure (OR: 2.29 (2.01, 2.59), $p < 0.05$) (Figure 3c), ischemic cardiac disease (OR: 1.70 (1.48, 1.95), $p < 0.05$) (Figure 3d), and CKD (OR: 3.24 (2.20, 4.79), $p < 0.05$) (Figure 3e) were associated with AKI event in COVID-19. No significant heterogeneity was observed in terms of ischemic cardiac disease, whereas the heart failure data showed low heterogeneity. Therefore, the fixed-effect pattern was chosen among these studies. In addition, the random-effect model was used in CKD, because the heterogeneity was high.

Eleven studies consisting of 10,319 patients showed a relationship between COPD and COVID-19 patients with AKI. Moreover, three studies reported that peripheral vascular disease was associated with AKI in COVID-19 patients. The random-effect model was used for meta-analysis because $I^2$ was >50%. The result finds that patients with COPD (OR: 1.86 (1.35, 2.57), $p < 0.05$) (Figure 3f) and peripheral vascular disease (OR: 2.34 (1.20, 4.56), $p < 0.05$) (Figure 3g) were at higher risk of AKI.

### 3.2.3. History Use of NSAID

Four studies evaluated the role of the chronic use of NSAIDs in COVID-19 patients with AKI. The data had low heterogeneity; hence, the fixed-effect model was applied. The meta-analysis revealed that the chronic use of this drug was a risk factor for AKI in COVID-19 (OR: 1.59 (1.29, 1.98), $p < 0.05$) (Figure 4).

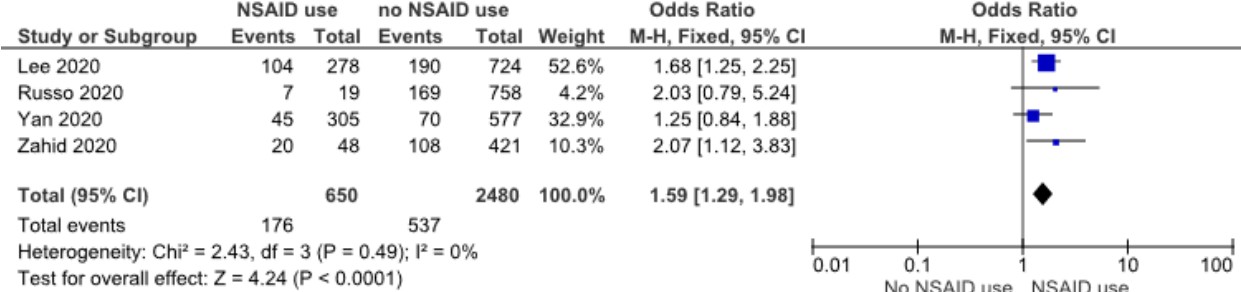

**Figure 4.** Forrest plot showing risk of AKI in COVID-19 with NSAID use [21,31,38,39].

### 3.2.4. Mechanical Ventilation Usage

A total of 9105 patients included in eight studies reported the relationship between mechanical ventilation use and AKI in 2019-nCoV disease. The heterogeneity test showed high heterogeneity, so we chose the random-effect model. The meta-analysis result revealed that the use of mechanical ventilation was associated with AKI events in COVID-19 patients (OR: 13.88 (8.23, 23.40), $p < 0.05$) (Figure 5).

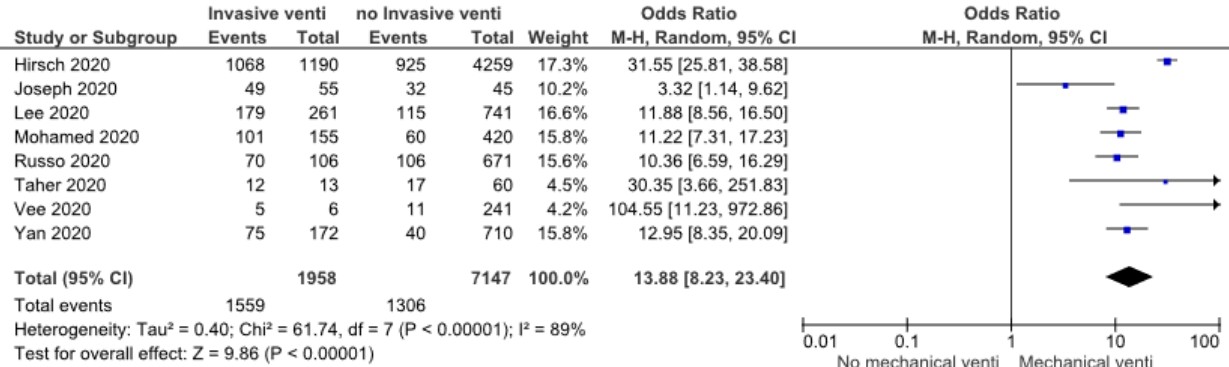

**Figure 5.** Forrest plot showing risk of AKI in COVID-19 with mechanical ventilation usage [18,20,21,25,31,33,34,38].

### 3.2.5. Proteinuria and Hematuria

The meta-analysis results found that AKI patients presented with proteinuria (OR: 3.31 (2.59, 4.23), $p < 0.05$) (Figure 6a) and hematuria (OR: 3.25 (2.59, 4.08), $p < 0.05$) (Figure 6b). We used a fixed-benefit model for these two parameters, because both of them had low heterogeneity.

Aside from the result, several variables did not show any relationship with acute kidney injury in SARS-CoV-2 infection in a meta-analysis. These factors were advanced age ($\geq$65 years old), obesity, smoking, cerebrovascular disease, cancer, chronic liver disease, HIV, and chronic use of aspirin (Figure S1). In addition, we evaluated the published bias for the included studies by funnel plot. The result showed that most of the studies showed no publication bias, implying that the data have been published with both positive and negative results.

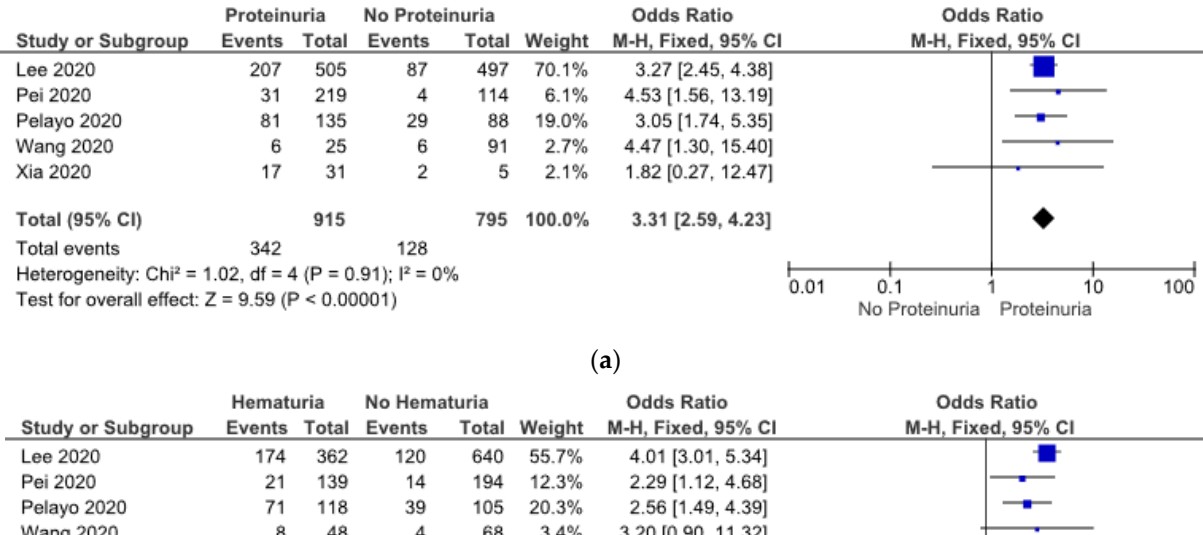

**Figure 6.** (**a**) Showing risk of AKI in COVID-19 patients with proteinuria, (**b**) showing risk of AKI in COVID-19 patients with hematuria [21,29,30,35,37].

## 4. Discussion

There are several risk factors that can put COVID-19 patients at higher risk of AKI. Our meta-analysis result found that AKI was associated with male gender, diabetes, and hypertension. Males were at higher risk, probably due to higher angiotensin-converting enzyme 2 (ACE2) expression and a less intense immune response [40,41]. Higher estrogen levels in females have a protective effect on the vascular endothelium [24]. The study involving 355 subjects showed that comorbidity with diabetes is an important independent risk factor predicting AKI among COVID-19 patients [42]. Besides, a study reported that hypertension patients were more likely to have inflammatory reactions, organ and tissue damage, and deterioration of the disease [43]. The relationship between diabetes and oxidative stress, atherosclerosis, and endothelial cell dysfunction has been established, as has the association between hypertension and those [44]. These might support the possible mechanism of AKI due to transient renal hypoperfusion and delayed recovery from ischemia–reperfusion injury [45]. Hypertension-mediated organ damage (HMOD) is defined as the structural or functional alteration of the arterial vasculature and/or the organs it supplies that is caused by elevated blood pressure, and end organs include the kidneys [46]. Since these studies were conducted in various countries, our findings may represent the global population. Therefore, we should increase awareness of AKI among male patients with diabetes and hypertension.

Patients with ischemic cardiac disease and heart failure showed signs of subclinical left ventricular impairment or reduced ejection fraction (EF) in severe cases, which may lead to renal hypoperfusion [8]. A meta-analysis including a total of 341 patients revealed that the standardized mean difference of cardiac troponin I (cTnI) levels was higher in those with severe COVID-19, suggesting a direct cardiac attack [47]. Our findings showed that AKI was more prevalent in COVID-19 patients with ischemic cardiac disease and heart failure. However, further studies are needed to determine whether this finding is secondary to COVID-19's direct cardiac involvement, sepsis, or a proinflammatory state.

Compared with COVID-19 patients with other underlying conditions, those with CKD were admitted to the ICU 12 times more frequently. Moreover, they also had a 9-fold increase in hospital admissions [48]. The risk of AKI in patients with kidney disease ranged from 1.9- to 4.4-fold higher depending on AKI severity [49]. The study design did not distinguish between pre-existing CKD and COVID-19-associated kidney injury. Our results found that patients with CKD were about three times more at risk of AKI progressing. CKD causes marked alterations in the immune system, such as persistent systemic inflammation and acquired immunosuppression, which may lead to clinical deterioration, including AKI [50,51]. Nevertheless, the actual impact of COVID-19 on kidney structure and function in patients with chronic kidney disease still requires further investigation.

We also find that COPD is related to AKI in COVID-19. Viral infections in COPD patients exacerbate the systemic inflammation and slow the recovery of reported symptoms [52,53]. We suggest that COVID-19 in patients with COPD perhaps may have a different inflammatory profile and should be considered distinct, and might contribute to AKI events. Moreover, the meta-analysis also revealed that peripheral vascular disease was associated with AKI in COVID-19. SARS-CoV-2 infection may be directly responsible for peripheral vascular disease, including venous and arterial thrombosis [54]. Global inflammatory response and endothelial damage were the proposed mechanisms; moreover, ACE2 expression was also abundant in the endothelial layer [55].

COVID-19 patients with a history of chronic use of NSAIDs were at higher risk of AKI. The prevalence of NSAID users in COVID-19 patients with AKI was revealed to be up to two times higher than that of non-AKI patients [31,39]. NSAID-induced AKI is associated with hypoperfusion due to vasoconstriction and acute interstitial nephritis (AIN) and has been found to be the major cause of drug-induced AKI. In 15% of all patients with unexplained AKI, the pathogenesis was due to AIN [56]. Several studies confirmed that discontinuation of NSAIDs proved beneficial especially in people whose risk of renal adverse effects is high [57,58]. However, we could not determine the exact etiology of AKI in COVID-19 patients with NSAIDs, whether due to NSAID-induced hypoperfusion or AIN.

As many as 40% of patients admitted to hospitals with COVID-19 have AKI. Accordingly, it commonly presents with dipstick-positive hematuria and mild proteinuria [18]. Hematuria was present in 48% of patients, and proteinuria has been seen in up to 60% of 1002 patients [21]. In another study, hematuria was present in 65%, and proteinuria was present in 77% of COVID-19 patients admitted for acute renal injury. A case series of kidney biopsies on 17 COVID-19 patients had previously been conducted, with AKI and heavy proteinuria being the most common indications for a biopsy. Among those with podocytopathy on kidney biopsy, 71% of patients demonstrated proteinuria [59]. Proteinuria is an indicator of the glomerular capillary wall alteration and its permeability and is thus a useful biomarker of the severity of the glomerular damage [60]. The pathophysiology of COVID-19-associated proteinuria and hematuria could be related to unspecific mechanisms but also to COVID-specific mechanisms, such as direct cellular injury resulting from podocyte damage or podocytopathy [61]. Proinflammatory cytokines conspire to elicit from endothelial cells a change from their homeostatic functions to those that can contribute to thrombosis and local tissue injury. Cytokines such as interleukin (IL)-1a and IL-1b, IL-6, and tumor necrosis factor (TNF)-alpha, among others, contribute critically to normal host defense, but when produced in excess, they can perturb all of the protective functions of the normal endothelium and potentiate pathological processes. The untrammeled production of proinflammatory cytokines contributes to a cytokine storm [62]. Viral injury to podocytes and renin–angiotensin–aldosterone system (RAAS) activation may have contributed to proteinuria. The accumulation of angiotensin II may be responsible for nephron endocytosis and increased glomerular permeability with proteinuria [63]. In our findings, either proteinuria or hematuria showed an independent association with AKI in COVID-19, although more direct investigations would be required to clarify this issue.

Recent investigations involving 81 COVID-19 patients admitted to the ICU discovered that 66 (44%) patients had AKI, with 44% of those having stage III AKI. All patients with AKI stage III had RRT. However, all patients who received RRT and survived their illness eventually fully recovered renal function and returned to their baseline levels [64]. This study revealed that AKI and RRT therapy are frequent in critically ill patients with COVID-19. Those receiving RRT may have a low risk of developing chronic kidney disease. Other meta-analyses, which involved 2401 patients in 15 articles, revealed that the most common complications of COVID-19 were acute respiratory distress syndrome (ARDS) (OR: 100.36 (64.44–156.32), $p < 0.05$) and shock (OR: 96.60 (23.80–392.14), $p < 0.05$) [65]. Among our findings on risk factors and comorbidities of AKI in COVID-19, we found that patients requiring mechanical ventilation support carried the highest risk (OR: 13.88 (8.23, 23.40), $p < 0.05$). These findings suggest that pre-renal factors, including hemodynamic instability, play a major role in AKI pathogenesis. Based on the findings of our meta-analysis, we suggest that the pathogenesis of acute kidney injury in COVID-19 is complex and multifactorial.

Although SARS-CoV-2 mainly targets the respiratory system, renal involvement, particularly AKI, was also discovered in COVID-19 patients. We found that in a large cohort of hospitalized patients at both tertiary and community hospitals, the rates of AKI were higher than those reported in previous studies. AKI has been reported in up to 35%–50% of COVID-19 patients [15,18]. The key mechanisms of acute kidney injury in COVID-19 remain unclear, whether direct kidney attack via ACE2, secondary to a global inflammatory state, hemodynamic instability, or concomitant nephrotoxic medication. Understanding the exact mechanism will have therapeutic implications. A recent study reported that SARS-CoV-2 NP antigen was accumulated in kidney tubules with severe acute tubular necrosis, but without evidence of glomerular pathology or tubule-interstitial lymphocyte infiltration. Recent human tissue RNA sequencing data have demonstrated that ACE2 expression in the kidney tubules is nearly 100-fold higher than in the lungs, suggesting the kidneys as a potential direct organ target [66]. It was known that ACE2 receptors are the major binding site for SARS-CoV-2 in host cells [67,68]. In contrast, a study in China also reported that they did not identify SARS-CoV-2 in any of the 72 urine samples using polymerase chain reaction [69]. There is no official treatment for COVID-19. Generally, the treatment for COVID-19 consists of antiviral, antibacterial, immunomodulatory, and anti-inflammatory. Each therapy has a different mechanism of action, and some drugs have nephrotoxic effects [70,71]. Chloroquine and hydroxychloroquine require special attention especially in patients with kidney dysfunction, because of their renal excretion [72]. Antiviral drugs, namely, lopinavir, ritonavir, and remdesivir, also have the potential for kidney injury [70,71]. There were several cases of renal injury in remdesivir users [73]. In addition to antivirals, intravenous immunoglobulin has a risk of proximal tubular injury [70,71].

Our meta-analysis has several limitations. COVID-19 is a new disease; given the rapid continuous expansion of the COVID-19 literature, many cohorts had relatively short follow-up periods and limitations in their descriptions of details, and there are new cohorts being reported continuously. It should be noted that the majority of studies did not clearly classify the data based on COVID-19 severity. The classification of severity is also different between studies. Hence, we cannot provide a subanalysis based on disease severity. Moreover, some articles included only severe cases. This might result in an overestimation of risk factors due to the inclusion of predominantly severe cases. For this reason, the gross variation of AKI incidence among studies and its association with disease severity cannot be equally compared. In addition, more information, particularly regarding risk factors and comorbidities, was not available to extract. We could not conduct subgroup analysis according to AKI grade, RRT requirement, or other outcomes due to the limited number of data points. Many studies also did not briefly compare risk factors and our desired outcome. The differences in study design, baseline characteristics, and standard laboratory measurements could result in unequal representation. The screening and diagnostic protocol of COVID-19 might also differ in each center due to limited

resources. Additionally, baseline values and the sensitivity and specificity of a laboratory test differ across studies, which may affect the outcome. This might affect the number of populations in the included studies, resulting in a less representative analysis. Shock also has a great impact on the development of AKI, either with or without a cytokine storm. Since the original studies did not provide the number of shock patients in both AKI and non-AKI groups, we cannot analyze the direct association between them.

## 5. Conclusions

AKI carries high morbidity and mortality in COVID-19. There are several risk factors that can put COVID-19 patients at higher risk of AKI, including male gender, diabetes, hypertension, ischemic cardiac disease, heart failure, CKD, COPD, peripheral vascular disease, and history of use of NSAIDs. These factors may have an association with AKI in non-COVID patients as well. Proteinuria, hematuria, and the use of mechanical ventilation are frequently found in AKI patients with SARS-CoV-2 infection, suggesting that these are important characteristics of AKI. The results of several variables might be less representative due to the small sample size of the included studies and the fact that they were conducted in a few countries. Therefore, further investigation is still recommended. Future observational multicenter studies based on the etiology of AKI in COVID-19 patients are required.

**Supplementary Materials:** The following supporting information can be downloaded at: https://www.mdpi.com/article/10.3390/pathophysiology30020020/s1, Figure S1. Forrest plot showing association between risk of AKI in patients with advanced age (≥65 years old) (a), obesity (b), smokers (c), cerebrovascular disease (d), cancer (e), chronic liver disease (f), HIV (g), and chronic use of aspirin (h).

**Author Contributions:** Conceptualization: A.A.H., V.A.G. and A.T.; data curation: A.A.H., V.A.G., F.R.I., R.A. and A.T.; formal analysis: A.A.H., V.A.G. and F.R.I.; methodology: A.A.H., R.A., D.M.H. and M.T.; writing—original draft: A.A.H., V.A.G., F.R.I. and R.A.; investigation: F.R.I., R.A. and D.M.H.; project administration: D.M.H.; validation: A.T. and M.T.; supervision: A.T. and M.T.; writing—review and editing: D.M.H., A.T. and M.T. All authors have read and agreed to the published version of the manuscript.

**Funding:** This research received no external funding.

**Institutional Review Board Statement:** Not applicable.

**Informed Consent Statement:** Not applicable.

**Data Availability Statement:** Not applicable.

**Conflicts of Interest:** The authors declare no conflict of interest.

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
