# Peer review of "Risk Factors and Clinical Characteristics of Acute Kidney Injury in Patients with COVID-19: A Systematic Review and Meta-Analysis"

_pathophysiology, doi:10.3390/pathophysiology30020020_

Round 1

Reviewer 1 Report

General comments

The grammar and the wording is good enough. There are not many grammar, spelling and punctuation errors.

Abstract

The abstract is concise, involving all the necessary information about the study

Introduction

-        The aim of the study is clearly stated in the Introduction

Methods

-        The methods used are presented in a very detailed way

Results

-        The results are presented in a very extensive way.

-        The tables are really helpful and necessary for the completion of the authors work.

-        The authors did a good job mentioning also the variables that did not show any relationship with AKI.

Discussion

-        The discussion is of great quality.

-        The authors mention that the key mechanism of acute kidney injury in COVID-19 remains unclear (line 251-252). It would be a great idea if the authors could suggest how future studies could define this and which methodology should they follow.

-        The authors inform the reader about the study limitations.

Conclusion

From the presented data, the conclusion is complete and represents the work that the authors did.

Author Response

Dear Editors,
Thank you for your offer to revise and resubmit our manuscript for consideration for publication in Pathophysiology.

Following this letter are the editor and reviewer comments with our response including how and where the text was modified. Changes made in the manuscripts are marked using track changes. The revision has been developed in consultation with all coauthors, and each authors has given approval to the final form of this revision.

Thank you for your consideration.

Sincerely,
Artaria Tjempakasari

Response to Reviewer 1 Comments

Point 1: The authors mention that the key mechanism of acute kidney injury in COVID-19 remains unclear (line 251-252). It would be a great idea if the authors could suggest how future studies could define this and which methodology should they follow.

Response 1: We have followed the reviewer's suggestions. (Line 335-336).

Reviewer 2 Report

In this meta-analysis, authors have shown that in COVID-19, AKI was associated with male, diabetes, and hypertension. The strength of this analysis is that it covered studies conducted in various parts of the world and thus may represent the global population. The authors have clearly explained the limitations as well. The analysis looks sufficient to address the outcome. 

However, this reviewer has a few suggestions-

-Please go through the discussion carefully and provide the appropriate references. For example, line 172.

-Does any of these study measure kidney injury molecules, NGAL or Kim-1?

-There are some recent studies where the outcome of critically ill COVID-19 patients with acute kidney injury has been shown. For example, "Lowe et al. BMC Nephrology. (2021) 22:92". Authors may like to discuss these publications.

-Please make lines 221-22 Clear

-It would be better to strengthen the statement (line 244-246), "according to....multifactorial".

Author Response

Dear Editors,

Thank you for your offer to revise and resubmit our manuscript for consideration for publication in Pathophysiology.

Following this letter are the editor and reviewer comments with our response including how and where the text was modified (Please see the attachment). Changes made in the manuscripts are marked using track changes. The revision has been developed in consultation with all coauthors, and each authors has given approval to the final form of this revision.

Thank you for your consideration.

Sincerely,

Artaria Tjempakasari

Reviewer 3 Report

I read with interest the article from Hidayat and colleagues. In this manuscript, the authors aimed to perform a systematic review and meta-analysis to identify risk factors for AKI development in COVID-19 patients. Since kidney injury is one of the main complications observed in the ICU for COVID-19 patients, and directly impacts on mortality, this review is relevant for the scientific community. Please find my considerations below.

Major comments:

- The 30 studies included for the meta-analysis should be mentioned in the Results section. Please include the references. I also suggest to include a column in the Figures with reference numbers for each study.

- Please consider to show the forest plots for the variables that were significant (age ≥65 years old, obesity, smoking, cerebrovascular disease, cancer, chronic liver disease, HIV, and chronic use of aspirin) as supplementary material.

Minor comments:

- Please correct “COVID-19 patients” in line 16.

- The abbreviations CKD (lines 12 and 42), COPD (line 21), NSAID (line 22), WHO (line 33), RAAS (line233) should be defined. Please check for others.

- I suggest that Figure 1 should be mentioned in the Results section.

- The subdivision of figures is not very clear. I suggest to separate the forest plots for better understanding. Also, the resolution of these figures should be improved.

- Please review the sentence in lines 188-190 for the better understanding.

Author Response

(The authors gave the same response as above.)
